# The Ancient History of Peptidyl Transferase Center Formation as Told by Conservation and Information Analyses

**DOI:** 10.3390/life10080134

**Published:** 2020-08-05

**Authors:** Francisco Prosdocimi, Gabriel S. Zamudio, Miryam Palacios-Pérez, Sávio Torres de Farias, Marco V. José

**Affiliations:** 1Laboratório de Biologia Teórica e de Sistemas, Instituto de Bioquímica Médica Leopoldo de Meis, Universidade Federal do Rio de Janeiro, Rio de Janeiro 21.941-902, Brazil; 2Theoretical Biology Group, Instituto de Investigaciones Biomédicas, Universidad Nacional Autónoma de México, Ciudad Universitaria, CDMX 04510, Mexico; gazaso92@gmail.com (G.S.Z.); mir.pape@iibiomedicas.unam.mx (M.P.-P.); 3Laboratório de Genética Evolutiva Paulo Leminsk, Departamento de Biologia Molecular, Universidade Federal da Paraíba, João Pessoa, Paraíba 58051-900, Brazil; stfarias@yahoo.com.br

**Keywords:** peptidyl transferase center, origin of life, 23S rRNA, proto-tRNA, emergence of biological systems

## Abstract

The peptidyl transferase center (PTC) is the catalytic center of the ribosome and forms part of the 23S ribosomal RNA. The PTC has been recognized as the earliest ribosomal part and its origins embodied the First Universal Common Ancestor (FUCA). The PTC is frequently assumed to be highly conserved along all living beings. In this work, we posed the following questions: (i) How many 100% conserved bases can be found in the PTC? (ii) Is it possible to identify clusters of informationally linked nucleotides along its sequence? (iii) Can we propose how the PTC was formed? (iv) How does sequence conservation reflect on the secondary and tertiary structures of the PTC? Aiming to answer these questions, all available complete sequences of 23S ribosomal RNA from Bacteria and Archaea deposited on GenBank database were downloaded. Using a sequence bait of 179 bp from the PTC of *Thermus termophilus*, we performed an optimum pairwise alignment to retrieve the PTC region from 1424 filtered 23S rRNA sequences. These PTC sequences were multiply aligned, and the conserved regions were assigned and observed along the primary, secondary, and tertiary structures. The PTC structure was observed to be more highly conserved close to the adenine located at the catalytical site. Clusters of interrelated, co-evolving nucleotides reinforce previous assumptions that the PTC was formed by the concatenation of proto-tRNAs and important residues responsible for its assembly were identified. The observed sequence variation does not seem to significantly affect the 3D structure of the PTC ribozyme.

## 1. Introduction

The peptidyl transferase center (PTC) is the catalytic center of the ribosome. Being a specific region of the larger ribosomal subunit, it is responsible for binding activated amino acids together and performing peptide elongation during protein synthesis. Since the early 1980s, Carl Woese and Harry Noller noticed that the essential mechanism underlying translation might be RNA based [1]. Nevertheless, it was only in 1992 that Noller and his collaborators found experimental evidence to support the idea that the PTC was indeed a ribozyme. They confirmed that the activity of peptidyl transferase is held by the ribosomal RNA after treating ribosomes with proteases without prejudice to the peptidic bond formation [2]. Four years later, Peter Lohse and Jack Szostak carried out the in vitro selection of ribozymes with the capability “to synthesize ester and amide linkages, as does the ribosomal peptidyl transferase” [3]. Further studies confirmed that the ribosomal peptidyl transferase reaction was performed by a region smaller than 200 base pairs located in the 23S ribosomal RNA of prokaryotes. Eukaryotes contain a similar PTC located in their 28S ribosomal RNA.

The PTC region has been considered crucial in the understanding about the origins of life. It has been described as the most significant trigger that engendered a mutualistic behavior between nucleic acids and peptides, allowing the emergence of biological systems [4,5,6]. Additionally, the proposal of a First Universal Common Ancestor (FUCA) departed from the contingent appearance of an ancient ribozyme capable of binding amino acids together [7]. The emergence of this proto-PTC is a prerequisite to couple a chemical symbiosis between RNAs and peptides that further evolved both to (i) become the large subunit of the ribosome by the principle of accretion and (ii) to allow the emergence of the genetic code. Although there remains controversy in the literature about whether the PTC is ancient or not [8,9,10,11], its importance cannot be challenged as it composes the central core of the decoding language of biology. The PTC is a versatile catalyst [12] that works as a turnstile for binding 20 different and very specific L-amino acids together to compose every cellular protein [13,14].

The origin and initial evolution of the PTC is a fertile field of debate and discussion in the scientific community. Some works indicated that the PTC was formed by a duplication of ancient forms of RNA once its structure was symmetric [15,16]. Other studies proposed the formation of the PTC by the junction of smaller RNAs, such as primitive tRNAs. Tamura [17] analyzed the secondary structure of the PTC and observed topological similarities with the secondary structure of transfer RNAs. In this sense, Caetano-Anollés and Sun [18] used structural analyses to provide evidence that tRNAs were older than ribosomes and were coopted to operate in the translation machinery. Farias and collaborators [19] analyzed sequence similarities between reconstructed ancestral sequences of tRNAs and the PTC. They verified an identity of 50.5% between a modern PTC and concatemers of ancestral tRNAs. Additionally, Root-Bernstein and Root-Bernstein [20] studied the similarity between tRNAs and rRNAs from *Escherichia*, observing several tRNA sequences found along its 23S rRNA sequence. They also suggested that the ribosomal RNA might have functioned as a primitive genome. Farias et al. [21] reconstructed a 3D structure of the PTC based on an ancestral sequence of tRNAs and observed a structural similarity of 92% when compared to the PTC of the bacteria *Thermus thermophilus*. Additionally, Demongeot and Seligmann [22] performed comparative studies between the secondary structure of both tRNAs and rRNAs and suggested that rRNAs were probably originated from tRNA molecules. Together, all these data make evident a scenario for the origin of life in which an evolutionary and chronological connection can be observed between these two essential components of the translation system: tRNAs and rRNAs.

Due to its remarkable relevance to biology and to the origins of life field, new studies that approach issues relating tRNAs and rRNAs are indispensable to better clarify how the initial organization of biological systems took place. Even when most works about ribosomal structure indicate that the PTC is highly conserved among all forms of life, we were unable to find conservation analyses of this particularly interesting region of the ribosome among the ancient domains of life. In addition, it seems important to analyze both the sequence and structure of the PTC in detail to gain insights about the emergence of biological systems. In this work, the following questions were posited: (i) Which exact nucleotides from the PTC are conserved among prokaryotes? (ii) How was the PTC probably formed? (iii) How can molecular modeling answer questions about the 3D structure conservation of the catalytic site of the ribosome? (iv) Are there co-evolving clusters of nucleotides that were invariant throughout the PTC’s evolution? Herein, we used comparative genomics and information theory to unravel patterns of information variation and nucleotide conservation among PTCs using all complete sequences of 23S rRNAs available in the GenBank database [23].

## 2. Material and Methods

### 2.1. Download of Complete 23S Ribosomal RNAs from Public Databases

All available sequences (complete) of 23S rRNA were retrieved from GenBank using the following search: “23s ribosomal RNA [All Fields] AND complete [All Fields] AND biomol_rrna [PROP]” with the nucleotide search function of the National Center for Biotechnology Information (NCBI) website. This search resulted in 1434 sequences downloaded from GenBank [23].

### 2.2. Retrieving PTCs from 23S Ribosomal RNA Sequences

A PTC sequence containing 179 bp from the bacteria *T. thermophilus* was obtained [19] and used as bait to retrieve the PTC region from the other 23S rRNA sequences obtained. The selection of PTC regions was performed using the optimal pairwise alignment tool Needleman–Wunsch [24]. Each of the 1434 23S rRNA sequences downloaded were aligned to the PTC of *T. thermophilus* using the needle script provided in the EMBOSS package [25]. An in-house needleParser.pl Perl script was developed to retrieve the start and end coordinates of the alignment and another script named get_RegionByCoordinate.pl was used to retrieve the exact PTC from the 23S rRNA sequences obtained.

### 2.3. Multiple Alignment, Filtering, and Production of PTC Datasets

The 1434 PTC regions were multiple aligned using ClustalW [26] software. After visual inspection of the alignment, we noticed 10 sequences particularly divergent from the others, presenting exceeded nucleotides and possibly representing annotation errors. These sequences were filtered to provide a PTC dataset in FASTA format containing 1424 high-quality PTC regions.

The whole PTC dataset was also separated into three different subsampled datasets according to taxonomic information. We analyzed the whole dataset and sequences from two ancient domains of life: Bacteria and Archaea. Five PTC sequences from eukaryotic organisms were discarded from further analyses as it has been found that eukaryotes compose a derived clade originated from the Lokiarchaeota group of archaea from the subphylum Asgard [27,28]. Each domain dataset was analyzed separately. The conservation analysis of nucleotides was performed using sequence alignments and the WebLogo tool [29] was used to generate pictures identifying the most conserved nucleotide residues. Manual curation was also performed in the alignments in order to allow the identification of 100% conserved nucleotides.

### 2.4. Information Theory Analysis of PTCs

For the information theory analysis, we considered the pseudometric variation of information. Pseudometrics differ from metrics because two different elements can be at distance zero. The variation of information measures the distance between two messages, X and Y. This metric is given by the formula V(X,Y) = H(X) + H(Y) − 2 I (X,Y), where H(X) stands for the entropy of the random variable X and I(X,Y) is the mutual information shared between the two random variables X and Y [30,31]; both measures are considered in bit units. For the analysis of PTC sequences, we deduced from the alignment the discrete distribution of nucleotides at each position. This procedure allowed us to determine the information distance of any two sites on any set of sequences of the same length [30,32]. If two positions are at an information distance of 0, the occurrences of nucleotides at these positions are strictly predictable, i.e., it is possible to determine one nucleotide from the other. Note that this fact holds in conserved positions but also in the case when the sites present some sort of linked variation. The variation of information allowed us to cluster sites according to the information distance between them. Particularly, an intra-cluster information distance of 0 provided well-defined, non-fuzzy clusters, on which nucleotides within single clusters perfectly co-varied.

### 2.5. Mapping Conservation into 2D and 3D PTC Structures

The 2D structure of the PTC was obtained from Ribovision [33]. We downloaded the SVG picture from the large subunit of *T. thermophilus* and cut the region previously identified as the PTC. The picture was edited by hand using image editors. Plus, a predicted secondary structure for the PTC of *T. thermophilus* (comprising 179 bp) was generated using the software RNAstructure [34] and visualized in the foRNA applet [35]. Both structures were colored according to the variation of information of the PTC alignment of all the 1425 sequences. Both the sequence and the 3D structure of *T. thermophilus* 23S rRNA (PDB ID 4v4i) were downloaded and manually edited to obtain the PTC region only.

### 2.6. 3D Modeling of the Different PTC Sequences and Structural Comparisons

Using the UGENE software [36], consensus sequences were obtained from each alignment file. The ModeRNA webserver [37] was used to perform template-based 3D modeling using the *T. thermophilus* PTC as a model to predict the 3D structure of the consensus sequences from the three datasets under analysis. The modeled structures were structurally aligned using the RNA-align software from the Zhang Lab suite [38]. Finally, we used Chimera [39] to visualize the structural comparisons.

## 3. Results

### 3.1. PTC Datasets: Production and Taxonomic Analysis

On 7 October 2019, the GenBank database contained 1434 complete sequences of 23S ribosomal RNAs. All these sequences were downloaded in FASTA format and aligned (using an optimal pairwise alignment tool) to the PTC of the bacteria *T. termophilus* to map a PTC sequence region containing 179 bp. Using in-house Perl scripts, we parsed the PTC alignment information and generated a file containing 1434 PTC sequences. After an initial round of multiple alignment of the PTC dataset, we visually identified ten sequences that seemed too divergent in the alignments. These anomalous PTC sequences were removed from further analyses as they possibly represented sequences with inaccurate genome annotation [40]. Therefore, a dataset containing 1424 PTC sequences in FASTA format was produced. This dataset presented five sequences from eukaryotes, 118 sequences from archaea and 1301 sequences from bacteria. Inside the bacterial clade, we observed that the major groups sampled were Proteobacteria (564 sequences), Firmicutes (237 sequences), and Actinobacteria (153 sequences); another 347 sequences were divided into 25 other clades. From the Archaea domain, 81 sequences came from Euarchaeota, 33 from Crenarchaeota, three from Thaumarchaeota, and one from Nanoarchaeota. The five sequences from eukaryotes came from the fungi species *Encephalitozoon intestinalis*. Table 1 summarizes the information about the sequences obtained.

### 3.2. Multiple Alignment and Sequence Conservation of PTC Datasets

A multiple alignment of each PTC dataset was performed and followed by the analysis of sequence conservation in the main datasets. Table 1 shows that all dataset alignments were 179 bp in length plus the number of gaps added by the alignment tool to optimize the sequence alignment. As expected, due to the presence of highly divergent sequences, the dataset PTC-all presented the highest number of gaps (10) and the fewest number of 100% conserved nucleotides (42). The highest number of positions 100% conserved was found in the Actinobacteria dataset, with 73.7% of conserved nucleotides and no gaps found. This was followed by Firmicutes, with 66.3% of residues completely conserved, and Proteobacteria, with 59.1%, evidencing a higher diversity of PTC sequences in the latter clade. This possibly happened due to the existence of large sub-clades (such as Alpha-, Beta, Gamma-, and Deltaproteobacteria), presenting well-differentiated PTC sequences among them.

The multiple alignments of the three main PTC datasets, i.e., PTC-all, PTC-Bac, and PTC-Arc, are displayed as sequence logos (Figure 1), on which the conservation profile can be easily visualized. In sequence logos, the 100% conserved positions were shaded in green, and the main adenine located at the catalytic site was starred on top. The observed single nucleotide gaps reveal variability in size among sequences. In some cases, the difference was detected in a single nucleotide present in one individual sequence, such as (i) the gap observed in position 21 of the PTC-all dataset; (ii) position 169 in PTC-all (and its equivalent position 165 in PTC-Bac) appeared merely due to the presence of a G in the bacterium *Spirochaeta africana*; and (iii) position 16 in PTC-Bac that was represented due to a C observed solely in the bacterium *Streptococcus pyogenes*. Other exceptional cases of gaps include positions 106 and 120 in PTC-all (equivalent to positions 101 and 115 in PTC-Bac) that correspond to single nucleotides observed in two Gammaproteobacteria species. In other cases, the differences rely on a few sequences that belong to one specific clade; for example, the gap in position 53 in PTC-all and position 49 in PTC-Bac correspond to a T present in Firmicutes species.

The lowest nucleotide conservation observed in the PTC sequences spans from nucleotide 37 to 44 and 65 to 70 in PTC-all and happened due to the fact that several bacteria from the phyla Chloroflexi, Aquificae, Fusobacteria, Bacteroidetes, Thermotogae, and Elusimicrobia (as well as some archaea) present divergent nucleotides in that region. In PTC-Bac, the sections ranging from 32 to 41 and 62 to 66 are observed to be more variable because the sequence data for the aforementioned phyla do not have analogous counterparts in archaea.

Considering the gaps spanning more than one nucleotide, it is possible to see a void at the beginning of the PTC-all alignment. It corresponds to a certain variation observed in the 5′ PTC region of some euryarchaeal sequences that seem to present a duplication in their five initial nucleotides. Such a region has a more notorious representation in the PTC-Arc dataset. The gap observed at the 3′ end of the PTC-all alignments and nearly at the end of the PTC-Bac alignments appeared due to one single sequence coming from the cyanobacterium *Thermosynechococcus elongatus* that differs from all other organisms. Therefore, the multiple alignment tool positioned the sequences in the most convenient way, either keeping the gap at the very end of the PTC-all dataset alignment (from nucleotides 194 to 201) or introducing a gap just before the end, as observed in PTC-Bac (between sites 181 and 185).

Interestingly, the only gaps observed in the PTC-Arc dataset appear at both ends (5′ and 3′). The initial one corresponds to the previously mentioned apparent duplication of the initial nucleotides in some euryarchaeal sequences. There is also a short portion at the last two nucleotides of the PTC-Arc dataset that shape an apparent gap at the end of alignment. This is observed because the PTC from *Nanoarchaeum equitans* and from some crenarchaeal sequences have a couple of G nucleotides inserted there. Additionally, the PTC-Arc sequence logo (Figure 1c) shows the highest amount of heterogeneity among all three datasets, observed in sites with different sizes of nucleotides along the vertical axis. Notably, even if the PTC-Arc dataset presented the highest number of conserved sites along the two domains analyzed (83 sites shaded in green for archaea compared to 62 for bacteria), its non-conserved sites also displayed higher variation. This fact denotes both the conservation of a tight PTC structure and a significant variability of this diverse clade that originated eukaryotic organisms [27,28].

Finally, site A2451 from the 23S rRNA is the catalytic site of the PTC, essential for the peptide bond to occur. This nucleotide has been shown to be absolutely preserved in each and every analyzed sequence, as it can be observed at the highlighted site with a magenta star at A17 in PTC-all, A13 in PTC-Bac, and A17 in PTC-Arc (Figure 1).

### 3.3. Mapping 100% Conserved Sites into the 3D Structure of the PTC

To gain insights about the nucleotide conservation in the PTC, we produced 3D models in which the 100% conserved positions (shaded green in Figure 1 and drawn in black in Figure 2) could be seen over the tridimensional structure. Thus, we obtained the PTC consensus sequences for each dataset and modeled their 3D structures using ModeRNA software using the known PTC structure for *T. thermophilus* (PDBid 4V4I) as a template. Analyzing Figure 2, we observe a higher number of conserved positions aggregated at the top of the structure (shaded oval in Figure 2), close to the catalytic site A2451 (identified with a magenta star). Nevertheless, the entire structure is significantly conserved, and specific conserved nucleotides located at different sites along the whole structure are probably anchors for holding the 3D shape of the PTC.

### 3.4. Identity Elements, Entropy Variation, Information Variation

For the complete set of sequence alignments, an entropy value was determined for each position. Thus, nucleotides were grouped into information clusters that were identified by color codes along the sequences (Appendix A). The first graph (Appendix A) corresponds to the analysis of both bacterial and archaeal sequence alignments, while Appendix A present the data for bacterial and archaeal sequences, respectively. Given that the number of sequences used for the informational analysis decreased between the PTC-all dataset and the archaeal one, the number of information clusters and positions in the clusters increased due to the reduction in variation. Ungapped positions, such as 45, 73, and 116 observed in the PTC-all dataset, have shown entropy close to the maximum value of two, meaning that all four nucleotides occurred in almost equiprobable amounts. Colored clusters with entropy equal to zero represent invariant nucleotide positions (Appendix A); while clusters with entropy greater than zero reflect positions on which nucleotide variations are highly predictable. This property can be clearly noticeable in the entropy profile for archaeal sequences (Appendix A) in which the positions in the red-colored cluster have entropy greater than 0. In each plot of Appendix A, the red color is associated with the modal cluster, i.e., the cluster containing the highest number of interdependent, co-evolving nucleotide positions.

The PTC-all dataset showed 25 nucleotide positions grouped into nine information clusters. Six of these clusters contained two positions and the remaining three clusters contained six, four, and three positions each. The information cluster harboring three positions was the only one in PTC-all whose bases were invariant, i.e., showed entropy equal to zero. The PTC-Bac dataset presented 71 positions divided into 11 clusters: Five clusters contained two positions, whereas the others presented 20, 11, 10, 8, 7, and 5 nucleotide positions. The cluster with the highest number of positions (20) was the only one that presented an intra-cluster entropy of zero. Regarding the PTC-Arc dataset, 101 positions were found split into 16 information clusters: 12 clusters contained two positions and the others possessed 47, 20, 7, and 3 positions. In the archaea data, the cluster containing 20 nucleotides was unique and contained an intra-cluster entropy equal to zero, representing invariant positions. All the nucleotide positions of the clusters are shown in Appendix A.

### 3.5. Mapping Identity Elements and Information Clusters into the 2D Structure of the PTC

A subtle variation was produced for each dataset alignment to consider only their sequence alignments without gaps, so that the length matched the canonical 179 nucleotides from the PTC. Even knowing that gaps are key, and their removal might produce artefacts, we proceeded to choose fixed length alignments to try to better explore the results, as the following analyses required these sorts of input data. Thus, information clusters were computed one more time and mapped into both (i) the known secondary structure of the PTC from *T. thermophilus* (Figure 3a) and (ii) the secondary structure predicted de novo by the software foRNA, using the PTC-all consensus sequence as entry (Figure 3b).

Analyzing Figure 3, one observes that the de novo structure differs from the modern PTC in two significant regions: (i) A stem formed on the first segment by the pairings of the bases from 9:25 (Figure 3, red star) up to 14:19 (Figure 3, blue star) and forming a loop with the segment 15–18 (Figure 3b) and (ii) the extra pairings of bases 75:159 and 76:158 (Figure 3, green star), on which the segment that joins the two coils in the secondary structure of the PTC can be seen. These three stars represent the main topological differences in the 2D structures (shown in Figure 3a,b), as all the other nucleotides are arranged similarly. The stars indicate which Watson–Crick base pairings should be broken in the lower entropy arrangement of Figure 3b to produce a modern-like PTC structure, as observed in Figure 3a. It is possible that these two RNA structures were viable and interchangeable at the origins of the PTC in prebiotic Earth, as it is known that RNA molecules can adopt different conformations [41,42,43]. These alternative foldings (among other possible ones) possibly changed according to the presence of different ligands and environmental conditions [44].

### 3.6. Mapping Proto-tRNAs into the 2D Structure of the PTC

When trying to explain how the PTC might have been formed in the past, we benefited from the work of Farias [45], who obtained putative ancestral tRNAs (Appendix A) using a dataset of 9758 sequences downloaded from a tRNA database [46]. In that work, Farias (2013) [45] separated 22 types of tRNAs, including 20 canonical tRNAs, one initiator tRNA, and one tRNA for selenocysteine, ran ModelTest to find the best nucleotide substitution model, and produced ancestral sequences using an approach based on maximum likelihood. In a following publication, Farias and collaborators (2014) [19] used a combinatorics approach to randomly concatenate those ancestral proto-tRNAs and search for possible matches in a nucleotide alignment against protein databases. Notably, these researchers found a specific combination of five proto-tRNAs concatenated directly (+/+ strands) that was shown to present 50% nucleotide identity to the PTC of the bacterium *T. thermophilus*. Therefore, to check whether the early origin of the PTC could be explained by the concatenation of those proto-tRNAs, we took these ancestral proto-tRNAs that bound to the amino acids (i) proline (Pro), (ii) tyrosine (Tyr), (iii) phenylalanine (Phe), (iv) glutamine (Gln), and (v) glycine (Gly) and aligned them to the PTC of *T. thermophilus*. These five proto-tRNAs were therefore mapped (in the order described above) within the segments 1–18, 19–54, 55–104, 105–149, and 150–179 of the modern PTC from *T. thermophilus* and plotted in colored boxes, as illustrated in Figure 3. As observed in the original publication (Farias, 2013) [45], the ancestral proto-tRNAs presented variable sizes (as measured in base pairs) due to the variable nucleotide conservation observed in the modern tRNAs used to produced them. Therefore, the maximum likelihood model, applied by Farias (2013) [45], removed some nucleotide positions that were not shown to be conserved in most tRNAs used to build the ancestral sequences and produced a sort of “truncated” ancestral proto-tRNA. The higher the nucleotide conservation in the modern tRNA sequences (to build the ancestral sequences), the longer the proto-tRNAs were.

We proceeded to analyze the co-occurrence of those proto-tRNAs and information clusters to check whether it could provide us with some insights. To do that, we started analyzing the PTC-Bac dataset due to the fact that it presented an intermediate number of clusters, as PTC-all contained too few positions in clusters (25 nucleotides), and PTC-Arc presented too many (101 nucleotides). Thus, analyzing the bacterial dataset (containing 71 nucleotides in 11 clusters), we noticed that four out of nine information clusters contained bases corresponding to the positions located in nucleotides placed in regions mapped in two different proto-tRNAs. In particular, Figure 3 provides evidence that (i) the cluster colored in yellow contained bases (black circles) putatively coming from proto-tRNA^Pro^ and proto-tRNA^Tyr^; (ii) the purple cluster contained bases (black circles) within proto-tRNA^Tyr^ and proto-tRNA^Phe^; (iii) the orange cluster contained bases (black circles) found in proto-tRNA^Phe^ and proto-tRNA^Gln^; and (iv) the dark blue cluster contained bases (black circles) found in both proto-tRNA^Gln^ and proto-tRNA^Gly^. We hypothesize that these information clusters represent co-evolving nucleotides originally responsible for linking the proto-tRNAs together in a higher-level secondary structure of extreme relevance to shaping the overall PTC structure. The cluster represented by (v), shown in red dots, contained bases mapped in all proto-tRNAs and was possibly relevant to the assembly of the whole 3D structure of the PTC. The one (vi) cluster shown with green dots contained a segment corresponding only to the third proto-tRNA^Phe^ (Figure 3). The other five clusters from the PTC-Bac dataset contained merely two bases representing Watson–Crick base pairs inside regions mapping single proto-tRNAs.

Similar data about the relationship between proto-tRNA mapping and information clusters are presented for PTC-all and PTC-Arc (Appendix A). Regarding the PTC-all dataset, we found one cluster containing six positions to have nucleotides coming from all the five proto-tRNAs. This finding reinforces the previous hypothesis of PTC formation by the concatenation of proto-tRNAs and suggests that this informational relationship might help to bind the proto-tRNAs together. Another identity cluster containing four positions was found in sites present in both proto-tRNA^Gln^ and proto-tRNA^Gly^. The PTC-all cluster, with three positions, embraced the first, fourth, and fifth proto-tRNAs and two clusters with two bases were found to present nucleotides in two regions mapped to different proto-tRNAs. Regarding the PTC-Arc dataset, the two clusters containing the highest number of nucleotide positions (47 and 20), encompass regions mapped in all the five proto-tRNAs. The PTC-Arc cluster, containing seven nucleotides, mapped into the third, fourth, and fifth proto-tRNAs. A cluster with three positions was mapped into the first, fourth, and fifth proto-tRNAs. Finally, six out of 12 clusters, containing two positions, mapped to two different proto-tRNAs. It is also conspicuous that in both the PTC-all and PTC-Arc datasets, there exist identity clusters containing nucleotide positions shared by non-consecutive proto-tRNAs. Altogether, those mappings reinforce the hypothesis that these information clusters account for the 3D configuration of the modern PTC by linking together ancestral proto-tRNAs.

### 3.7. Mapping Identity Elements and Information Clusters into the 3D Structure of the PTC

In Figure 4, the tridimensional structure of the PTC derived from *T. thermophilus* 23S rRNA is shown and colored according to the different information clusters found in the PTC-Bac dataset, aiming to observe their spatial distribution. For all datasets, the most evident characteristic is that many identity clusters bundle in a tridimensional configuration (data for PTC-all and PTC-Arc are shown in S3). Besides the modal cluster (in red), most other nucleotides sharing similar clusters are observed in nearby positions. Brown sites, for example, remain together in the arm observed at the right side of the structure in Figure 4. Most of the green and dark blue nucleotides are close together in the 3D shape, near the yellow cluster that contains the catalytic site (A12 in this structure, marked with a magenta star). The red-colored nucleotides—corresponding to the cluster containing a higher number of interrelated, co-evolving positions—are spread throughout the PTC molecule, possibly responsible for maintaining the whole structure. They are absent, however, from the bottom of one arm (observed at the left side of Figure 4a) in which purple, light blue, and pink clusters mostly muster. Therefore, all clusters seem necessary to maintain the PTC structure and create the cage necessary for the peptide bonds to occur.

### 3.8. Structural Alignment of PTC Datasets to T. Thermophilus

The template-based structural reconstructions for the consensus sequences from the PTC-all, PTC-Bac, and PTC-Arc datasets were structurally compared to the actual PTC known for *T. thermophilus* in order to provide an appreciation of the spatial differences among them (Figure 5; in which cyan-colored regions represent a match between predicted and actual PTC structures and purple represents differences).

As expected, due to the high general conservation shared among their sequences, the template-based structures were similar to the PTC of *T. thermophilus* [47], while the most relevant variations were observed towards their extremities. Indeed, the most similar structure to *T. thermophilus’* PTC was PTC-Bac, which was shown to be only slightly longer than the model at the 3′ end of the structure. The most patent difference between the model and the consensus of PTC-all was the presumptive insertion of five nucleotides at the 3′ end, whereas it was arranged as an internal insertion positioned at the top of the PTC-Arc structure.

In the portions corresponding to nucleotides 116 to 123 and 166 to 173 of the PTC from *T. thermophilus*, both PTC-all and PTC-Bac consensuses differed from the model. However, the alignments revealed only a slight difference in the corresponding segments. An opposite situation happened in the comparison to the PTC-Arc structure, in which the sites 116 to 119 revealed a highly variable sequence region though the consensus structure and the models were shown to be nearly undistinguishable.

## 4. Discussion

The sequence of the PTC is possibly the most relevant stretch of nucleic acid to be studied if one aims to understand the origin of life. Nowadays, it is a consensus that the ribosome should be understood as a prebiotic machine that predated the origin of cells [16,48]. Although there is a debate in the literature about whether the ribosome was built over the PTC region or not [9,10,11], many researchers claim to have evidence that the assembly of the genetic code and the ribosome started with the initial formation of region V of the ribosome, just the place in which the PTC is settled. Theoretical works on the origin of life suggest that the contingent appearance of this ribozyme capable of binding amino acids together was crucial to both the initial emergence and further development of the phenomenon of life [7,49,50].

Here, we evaluated the sequence and structural conservation of the peptidyl transferase center using all completely available sequences of 23S ribosomal RNA present in the GenBank database and annotated as such. We decided to use complete sequences to add rigor to the analyses and to avoid to the sequencing errors often present in small molecules [40]. Besides, as we were interested in understanding the relevance of the PTC to the early origin of life, we decided to exclude eukaryotic sequences from the analyses. Eukaryotes are now known to have originated from archaeal organisms coming from the phylum Lokiarchaeota, subphylum Asgard [27,28], therefore being derivate clades and having no substantial role in early origins of life.

The 1424 23S rRNA sequences obtained were aligned, filtered out to retrieve only the PTC region, and divided into three main datasets. Although there is a consensus in the literature about the fact that the PTC sequence is highly conserved [51,52,53,54,55], to our knowledge, PTC sequences have not been analyzed thoroughly by comparative sequence analysis, information variation, and bi/tri-dimensional structural analysis to better validate this assumption. 

The multiple alignment comparison showed that the PTC from archaea presents about 40% fully conserved bases; this number lowers to 30% in bacteria, and to 20% in all analyzed organisms (Table 1). These percentages of conservation are clearly dependent on the total number of sequences analyzed, as bacteria possessed >11x more sequences available in GenBank than archaea (1302 versus 118). Curiously, the archaeal dataset presented both a higher number of conserved nucleotides (83) and a higher number of variable sites when compared to the bacterial one. This possibly means that the structure of the archaeal PTC is more optimized to be tighter in specific regions that maintain a rigid tridimensional backbone and looser in others. By contrast, the structure of the bacterial PTC is possibly wobblier.

When observed in the context of the bi- and tridimensional structures, we found that most fully conserved bases from the PTC folded close to the catalytic site, whereas sites located down to the two hairpin structures seem to allow more variation (Figure 2). This last fact should be expected, as the catalytic sites of enzymes are often more conserved than the other parts; the same is true for ribozymes. The PTC is known to be a flexible and efficient catalyst [12] as it is capable of recognizing different, specific substrates (20 different amino acids bind to aminoacyl-tRNAs) and polymerizing proteins at a similar rate [56,57]. Therefore, considering the extreme relevance of the PTC, it would be surprising if the site of catalysis showed variation.

The use of a pseudometric to show the information variation in PTC sequences allowed us to identify clusters of nucleotides that are informationally linked. We were able to find clusters containing as many as 47 nucleotides, although most of them presented fewer than 10 nucleotides. The clusters with a higher number of bases were colored in red for all datasets and they were invariant for PTC-all and PTC-Bac, although in PTC-Arc, this modal cluster presented 47 nucleotide positions that could vary coordinately. We hypothesize that these clusters were mainly important to keep the tridimensional structure of the PTC, but we found out that they also provided interesting insights about how the PTC was formed.

Farias and collaborators [19] used tRNA sequences from hundreds of species, together with maximum likelihood analyses, to construct ancestral sequences for each of the 20 different tRNAs, producing putative ancestral proto-tRNAs. When they randomly concatenated these proto-tRNAs and BLASTed them with GenBank’s nucleotide database, they verified that one concatemer of five proto-tRNAs presented a significant nucleotide identity (about 50%) to the 23S rRNA of the bacterium *T. thermophilus,* exactly in the PTC region [19]. Even if 50% identity cannot be considered a significant threshold for sequence identity, one cannot expect to apply modern standard measures of nucleotidic variation when working with an event so distant in the past. Therefore, even considering the hypothetical nature of this result, we decided to go further into that investigation. Thus, we mapped their five proto-tRNAs concatamers into the 2D structure of the *T. termophilus* PTC. Besides mapping the proto-tRNAs, we also produced a diagram in which the nucleotides were colored according to the clusters produced with the information variation analysis. This resulted in Figure 3a, which showed the 2D structure of the PTC with dots representing each nucleotide of the PTC colored according to its corresponding cluster. We found out that many information clusters contained informationally linked nucleotides mapped to distinct proto-tRNAs along the PTC structure. This fact seems to indicate that these nucleotides may have been relevant to the stepwise binding of these ancestral tRNAs with each other in order to produce the modern shape of the PTC site. These results are in accordance with recent works suggesting that either the PTC or rRNAs should have been formed by the assembly of tRNAs [17,18,20,22]. We not only confirmed these previous assumptions but added new information on the conserved sites possibly used to link the PTC structure. Additionally, we used a de novo modeling software to predict the 2D structure of the PTC (Figure 3b) and were able to produce a structure very similar to the modern PTC. Working over this structure, we were able to identify four sites linked by Watson–Crick bonds that, when released, may have given rise to the modern PTC (these sites were identified by three colored stars in Figure 3). Our hypothesis is that the ancient PTC could be observed in at least these two structures, changing from one to the other according to the presence of ligands and specific environmental conditions—a known property of RNAs—to present multiple unstable, interchangeable structures [41,44]. Additionally, a hairpin with yellow dots observed in the bottom part of the de novo structure (Figure 3b) clearly indicated which nucleotides were used to bind proto-tRNA^Pro^ and proto-tRNA^Tyr^ into an integrated, higher-level structure that produced the PTC. Although the binding of other nucleotides between distinct proto-tRNAs cannot be clearly observed in the current structure of the PTC, we hypothesize that these co-evolving nucleotides were important to bind the proto-tRNAs together when the PTC was under formation, as its secondary structure probably grew by the addition of one tRNA at time. Thus, the informationally linked nucleotides possibly held the higher-level 2D and 3D structures together to allow the stepwise formation of the whole PTC region.

As both the position of nucleotides sharing the same cluster (Appendix A) and the bidimensional structure of the PTC (Figure 3) looked sometimes distant and peculiar, we decided to gain new insights by plotting the clusters into a tridimensional structure. We used the same color code for clusters to check whether the position of the nucleotides sharing same clusters would make more sense in 3D and we found that this was indeed the case for most clusters (Figure 4). Both the ribbon and the surface structures demonstrate that nucleotides sharing the same information cluster were usually observed close to each other in the 3D structure.

An interesting possibility derived from the current analyses would be the actual production of resurrection experiments [58] able to synthesize the putative form of an ancient PTC using the exact nucleotide sequence derived from the concatamer of proto-tRNAs described here. Which properties may this molecule have? How would it fold? Could this sequence function as a ribozyme and catalyze a peptide bond? Similarly, it could be possible to synthesize the proto-tRNAs proposed by Farias (2013) [45] and verify whether they could bind with each other using the nucleotides described in the information variation model we used. Those experiments could bring an experimental background to the theoretical analyses performed here. 

Finally, the predicted 3D structures based in the consensus sequences for each dataset were structurally compared to the known PTC structure from *T. thermophilus* [47] to allow for the identification of similarities and divergences. Despite some notable nucleotide variations from PTC-all and PTC-Bac to the *T. thermophilus* sequence, the 3D structure was shown to be significantly conserved. The higher divergences found were to be related to slight extensions observed in the 3′ regions of the datasets. Regions containing nucleotide variance were shown to be conserved at the structural level (Figure 5). In conclusion, we have provided (i) a better understanding of how nucleotide variation is observed in the PTC, underscoring (ii) a testable possible model about how proto-tRNAs shaped its structure, and (iii) how the evolutionary process froze essential nucleotide positions that enabled the peptide polymerization bonding by preserving the tridimensional structure of the PTC ribozyme.

## Figures and Tables

**Figure 1 life-10-00134-f001:**
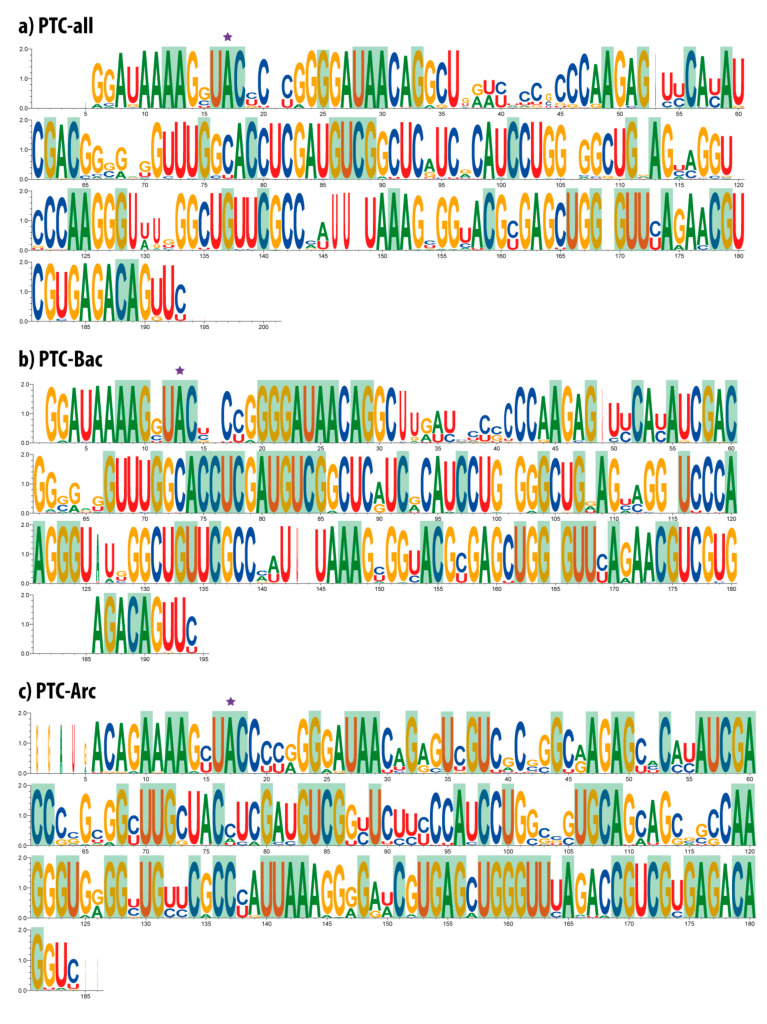
WebLogos showing nucleotide conservation in the main analyzed PTC datasets. (**a**) PTC-all; (**b**) PTC-Bac: (**c**) PTC-Arc. Universally conserved nucleotides in each dataset are shown with a green background. The adenine located at the catalytic site is highlighted with a magenta star.

**Figure 2 life-10-00134-f002:**
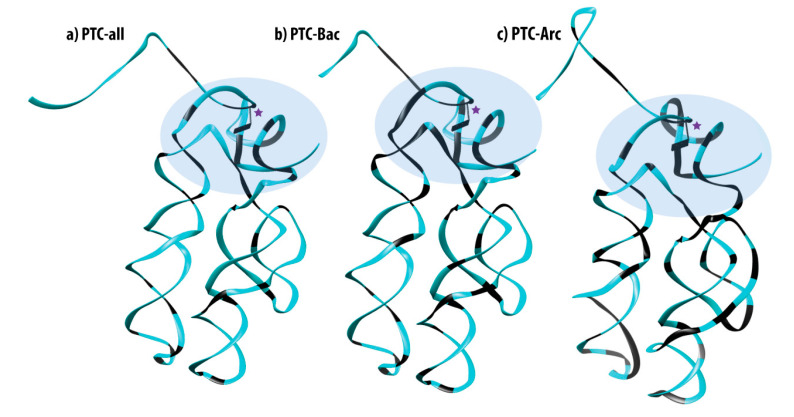
Embedding the 100% conserved nucleotides for each dataset: (**a**) PTC-all, (**b**) PTC-Bac, and (**c**) PTC-Arc. The structures were produced by template-based modeling over the known *T. thermophilus* structure. The catalytic site A2451 is depicted as a rectangle protrusion close to a magenta star. The 100% conserved nucleotides are colored in black and ovals accentuate the regions with highly conserved sites at the top of the structures.

**Figure 3 life-10-00134-f003:**
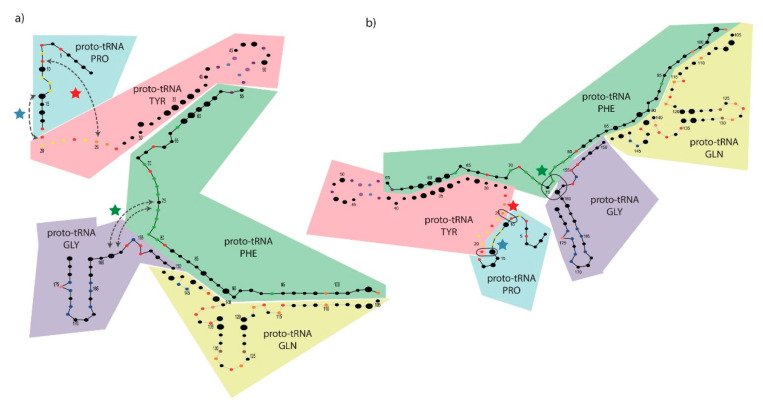
Information clusters and proto-tRNA composition from the PTC-Bac dataset as observed over (**a**) the secondary structure of the modern PTC from *T. thermophilus* and (**b**) the de novo (predicted) RNA structure. The radius of each circle corresponds to its entropy value, i.e., bigger circles represent more variable positions. Regions corresponding to each proto-tRNA are shown in colored boxes according to the corresponding ancestors. Colored stars in blue, red, and green represent Watson–Crick base pairing in putative ancestral folding (**b**) that were separated to generate the modern folding (**a**). Arrows in (**a**) represent the positions linked by Watson–Crick base pairing in (**b**) that were separated to produce the catalytic structure of the PTC (**a**).

**Figure 4 life-10-00134-f004:**
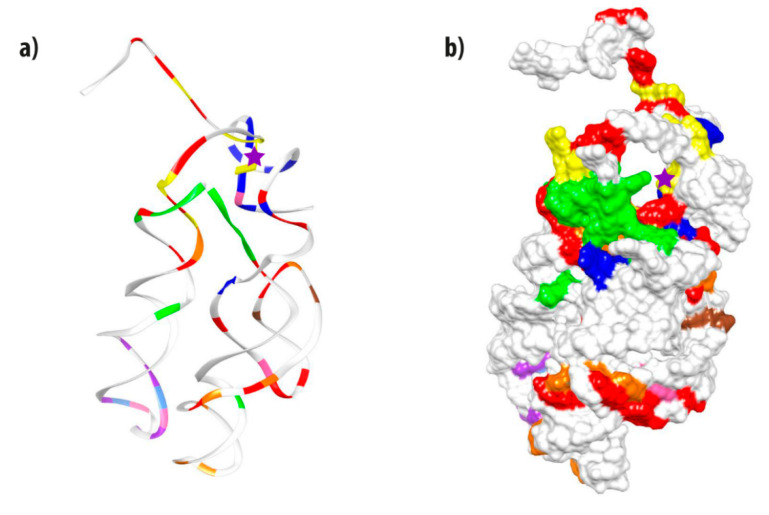
Ribbon (**a**) and surface (**b**) tridimensional representations of the PTC from *T. thermophilus* with the information clusters found in the PTC-Bac dataset colored according to Figure 3. The catalytic site (corresponding to A2451 or A12 in the current model) is marked with a magenta star and protrudes from the ribbon sketch (**a**).

**Figure 5 life-10-00134-f005:**
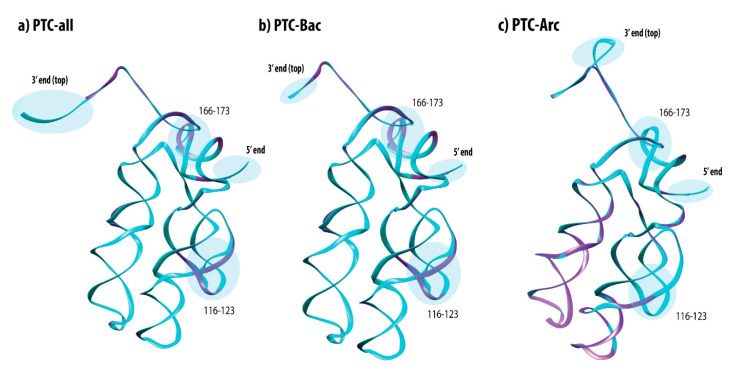
Tridimensional structural comparisons between the template PTC structures from *T. thermophilus* are shown in purple and the predicted PTCs, depicted in cyan for the different clades: (**a**) PTC-all, (**b**) PTC-Bac, and(**c**) PTC-Arc. Relevant regions are encircled and labeled.

**Table 1 life-10-00134-t001:** Peptidyl transferase center (PTC) sequences per clade, number of sequences, and multiple alignment features.

Dataset Name	Clades	Number of PTC Sequences	Alignment Size (Gaps)	Positions 100% Conserved
PTC-all	Bacteria Archaea Eukarya	1424	201 (10)	42
PTC-Bac	Bacteria *	1301	195 (8)	62
PTC-Arc	Archaea *	118	186 (7)	83
PTC-Pro	Proteobateria *	564	186 (7)	110
PTC-Fir	Firmicutes **	237	184 (4)	122
PTC-Act	Actinobacteria	153	179 (0)	132

* These datasets are subsets of the PTC-all dataset. ** These datasets are subsets of the PTC-Bac dataset.

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
