# Peer review of "The Ancient History of Peptidyl Transferase Center Formation as Told by Conservation and Information Analyses"

_life, 2020, doi:10.3390/life10080134_

Round 1

Reviewer 1 Report

I have only one suggestion for the authors, which is that they add a section to the Conclusion discussing possible experimental tests of their work. For example, modern technology should make it possible to synthesize a functional proto-PTC of 179 bp.  Similarly, it should be possible to ligate the five proto-tRNAs to make a proto-PTC. Etc. The analyses offered by the authors are excellent, but we must remember that they are still hypotheses and it is time for the field to move beyond theory to experiments!

Author Response

Reviewer 1

We thank reviewer 1 for considering appropriate both our manuscript and our English usage.

> I have only one suggestion for the authors, which is that they add a section to the Conclusion discussing possible experimental tests of their work. For example, modern technology should make it possible to synthesize a functional proto-PTC of 179 bp.  Similarly, it should be possible to ligate the five proto-tRNAs to make a proto-PTC. Etc.

We do really appreciate the suggestion of adding a section in the Conclusion proposing a resurrection experiment of our proto-PTC. The following paragraph has been added in Discussion:

“An interesting possibility derived from the current analyses would be the actual production of resurrection experiments (Zaucha and Heddle, 2017) capable to synthesize the putative form of an ancient PTC using the exact nucleotide sequence derived from the concatamer of proto-tRNAs described here. Which properties this molecule may have? How it would fold? Could this sequence function as a ribozyme and catalyze a peptide bond? Similarly, it could be possible to synthesize the proto-tRNAs proposed by Farias (2013) and verify whether they could bind with each other using the nucleotides described in the information variation model we used. Those experiments could bring an empirical basis for the theoretical analyses performed here.”

> The analyses offered by the authors are excellent, but we must remember that they are still hypotheses and it is time for the field to move beyond theory to experiments!

We really thank the reviewer to consider our analyses excellent.

The reviewer is right to remember that our proposal is still hypothetical and this specific word “hypothetical” was used in the last sentence of the manuscript to emphasize this aspect of our work.

Reviewer 2 Report

This is a potentially interesting paper on an interesting topic, but the way the paper is presented is not good. The authors must improve and clarify the presentation.

If the peptidyl transferase center (PTC) evolved from tRNAs, this raises the following issues. First, are these full-length tRNAs? How do you identify which tRNA is which? Are these correct identifications or possible identifications? If the PTC is made from truncated tRNAs, does a model arise for how the sequences were truncated?

Archaeal tRNAs are older than bacterial tRNAs. For instance, Pyrococcus furiosis tRNAs are very close to LUCA tRNAs. Pfu tRNAs might make a better search set for the PTC model.

Because bacterial and archaeal PTC are so similar with such similar features, what does this indicate about the divergence of Bacteria and Archaea?

Is there an indication of whether complementary replication occurred before evolution of the PTC?

I think this paper should be significantly improved before publication.   

  1. The authors indicate that the peptidyl transferase center (PTC) is generated from a concatemer of tRNA-like sequences. They do not indicate how these sequences match up based on tRNA motifs. They do not indicate whether these are +/+ or +/- alignments to tRNAs, which is an important issue for a reader and, also, to understand evolution of the PTC. Because of the lengths of the alignments, these must be truncated tRNAs. Are alignments to microhelices, minihelices or full length tRNAs?
  2. The authors should add a figure that gives best alignments of the PTC to tRNA segments with PTC features and tRNA features annotated. It is very difficult to refer to the earlier paper from this group to try to make sense of this paper.
  3. The segment of the PTC that is analyzed has many features that should be pointed out. For instance, H86, H87, H82, H88, A2451 and G2553 (Thermus thermophilus numbering). These features should be annotated. Otherwise, the importance of the PTC segment analyzed is not clear to the reader.
  4. This reviewer was dissatisfied with the primordial tRNA segment sequences indicated in the earlier paper by this group. The authors ignore and do not reference better information about primordial tRNA sequences [1-4]. These sequences may be suitable to the current purpose, but, without annotations, their suitability cannot be evaluated.  

References:

  1. Burton, Z.F. The 3-Minihelix tRNA Evolution Theorem. J Mol Evol 2020, 10.1007/s00239-020-09928-2, doi:10.1007/s00239-020-09928-2.
  2. Pak, D.; Du, N.; Kim, Y.; Sun, Y.; Burton, Z.F. Rooted tRNAomes and evolution of the genetic code. Transcription 2018, 10.1080/21541264.2018.1429837, doi:10.1080/21541264.2018.1429837.
  3. Kim, Y.; Kowiatek, B.; Opron, K.; Burton, Z.F. Type-II tRNAs and Evolution of Translation Systems and the Genetic Code. Int J Mol Sci 2018, 19, doi:10.3390/ijms19103275.
  4. Pak, D.; Root-Bernstein, R.; Burton, Z.F. tRNA structure and evolution and standardization to the three nucleotide genetic code. Transcription 2017, 8, 205-219, doi:10.1080/21541264.2017.1318811.

Author Response

Reviewer 2

We thank reviewer 2 for his/her criticisms. Even if (s)he did not point any specific problems in English usage, we have passed the entire manuscript for a new round of language review and did our best to rewrite it as clear as possible.

Overall, the reviewer is right to ask for more information about how the ancestral proto-tRNAs were built. To answer this criticism, we added a significant amount of information in section 2.6 and provide a further citation of the original manuscript on which they were produced (Farias, 2013; https://doi.org/10.1016/j.jtbi.2013.06.033). We hope the reviewer will be satisfied with this clarification and we really thank him/her for noticing this issue.

> This is a potentially interesting paper on an interesting topic, but the way the paper is presented is not good. The authors must improve and clarify the presentation.

We thank the reviewer to consider our paper potentially interesting. We have improved and clarified the presentation.

> If the peptidyl transferase center (PTC) evolved from tRNAs, this raises the following issues. [a] First, are these full-length tRNAs? [b] How do you identify which tRNA is which? [c] Are these correct identifications or possible identifications? [d] If the PTC is made from truncated tRNAs, does a model arise for how the sequences were truncated?

The suggestion that PTC evolved from tRNA has been already proposed earlier, please check the third paragraph of Introduction (page 4) and here we re-evaluate this possibility trying to add new information on how this evolution might have happened.

We would like to invite the reviewer to read the new version of section 2.6 in our manuscript, on which we tried to answer each of the following criticisms made by him (pages 19 and 20).

[a] The proto-tRNAs we are working are not full-length tRNAs. They have been produced in an earlier publication (Farias, 2013) on which tRNAs from prokaryotes were downloaded and subjected to a maximum likelihood (ML) analyses aiming to produce ancestral versions of these molecules.

[b] In that publication, Farias and collaborators took all the 22 tRNAs separately and made one ancestral for each of those. This way we know which tRNA is which and [c] these are correct identifications.

[d] The truncation of tRNAs comes from the fact that the ML method for ancestral reconstruction removes from the ancestral proto-tRNAs the nucleotides that are not conserved in all the modern tRNAs used as input for the analysis. For each tRNA, the algorithm provides a nucleotidic alignment, finds the best model of nucleotide substitution using ModelTest and runs the ML analysis in order to produce the most likely ancestral sequence.

> Archaeal tRNAs are older than bacterial tRNAs. For instance, Pyrococcus furiosis tRNAs are very close to LUCA tRNAs. Pfu tRNAs might make a better search set for the PTC model.

The reviewer is partially right and the question whether Archaeal tRNAs are older than Bacterial ones is controversial. In any case, the investigation we propose in the current manuscript aims to provide information on an event that happened much earlier than the formation of LUCA, i. e., before the evolution of cells. Therefore, the ML model that built the proto-tRNAs by Farias (2013) took on account both archaeal and bacterial sequences. The built proto-tRNAs therefore considered positions that were conserved in both domains and putatively represent proto-tRNAs existing before the origin of both Archaea and Bacteria. Also, the ancient PTC we propose here should be consider having existed before any form of cellular organism, at the dawn of the progenotes’ time. Finally, the choice to use the PTC of T. thermophilus as a model was given by the fact that it was the most similar sequence to the concatamer of 5 proto-tRNAs produced (please refer to Farias et al., 2014).

> Because bacterial and archaeal PTC are so similar with such similar features, what does this indicate about the divergence of Bacteria and Archaea?

Our Figure 2a shows clearly that nucleotide conservation is similar between all the sequences analyzed, but Figure 2b and 2c identify exactly which nucleotides are conserved either in Bacteria or Archaea. As commented in Results (please check the three last sentences on page 27; at the fourth paragraph of Conclusion), this means that archaeal PTC presents more conserved sites and also more variable sites. Therefore, archaeal PTC is more rigid at some specific points and more flexible in others. These points of rigidness and flexibility were observed for the first time and precisely identified. On the other side, the PTC from Bacteria shows more variability, presenting an overall structure that is less conserved. This possibly reflect the fact that Bacteria clade is more diverse than Archaea, but we cannot be confident about this once the number of sequences available from Genbank for Archaea is about 11x smaller than the number of sequences available for Bacteria.

> Is there an indication of whether complementary replication occurred before evolution of the PTC? I think this paper should be significantly improved before publication.

No. Even if we tend to consider that complementary replication occurred very early, being probably the most important form of RNA replication, our results are not compatible witj replications/duplications events.

> 1. [a] The authors indicate that the peptidyl transferase center (PTC) is generated from a concatemer of tRNA-like sequences. [b] They do not indicate how these sequences match up based on tRNA motifs. They do not indicate whether these are +/+ or +/- alignments to tRNAs, which is an important issue for a reader and, also, to understand evolution of the PTC. [c] Because of the lengths of the alignments, these must be truncated tRNAs. Are alignments to microhelices, minihelices or full length tRNAs?

Again, those criticisms were mostly answered by the addition of new information at the section 2.6.

[1a] The reviewer is right to say that our PTC has been generated by a concatamer of tRNA-like sequences. We name these sequences proto-tRNA and they were derived from the work of Farias (2013) (doi:10.1016/j.jtbi.2013.06.033).

[1b] We briefly describe how these sequences match up. Basically, Farias and collaborators (2014) performed concatenations of these 22 ancestral proto-tRNAs in all possible combinations (with concatenations of 3, 4 and 5 different proto-tRNAs) using both +/+ and +/- alignments. All this dataset was aligned against a PTC from multiple prokaryotes and they found (as their best result) that the concatenation of 5 specific tRNAs used here showed 50% identity to the PTC of the bacteria Thermus thermophilus. Once again, that is why we used T. thermophilus’ PTC in the current analysis.

[1c] The reviewer is right to say that these proto-tRNAs are variable in size, as measured in base pairs. These are ancestral RNAs that probably worked as tRNAs. They were probably “truncated” when compared to modern tRNAs, even if they probably functioned perfectly in their task to bring amino acids to the proto-ribosome.

> 2. The authors should add a figure that gives best alignments of the PTC to tRNA segments with PTC features and tRNA features annotated. It is very difficult to refer to the earlier paper from this group to try to make sense of this paper.

The reviewer is right, and we hope to have answered this criticism with the new version of section 2.6. This section describes better all the relevant information on our earlier papers to allow the readers to make sense of the current manuscript.

> 3. The segment of the PTC that is analyzed has many features that should be pointed out. For instance, H86, H87, H82, H88, A2451 and G2553 (Thermus thermophilus numbering). These features should be annotated. Otherwise, the importance of the PTC segment analyzed is not clear to the reader.

The goal of Figure 4 was to compare the modern structure of PTC of T. thermophilus with our cluster analysis and with our proposed proto-PTC (Figure 3). Adding the features suggested by the reviewer is beyond the scope of our main goal in this section.

> 4. This reviewer was dissatisfied with the primordial tRNA segment sequences indicated in the earlier paper by this group. The authors ignore and do not reference better information about primordial tRNA sequences [1-4]. These sequences may be suitable to the current purpose, but, without annotations, their suitability cannot be evaluated.  

References:

Burton, Z.F. The 3-Minihelix tRNA Evolution Theorem. J Mol Evol 2020, 10.1007/s00239-020-09928-2, doi:10.1007/s00239-020-09928-2.

Pak, D.; Du, N.; Kim, Y.; Sun, Y.; Burton, Z.F. Rooted tRNAomes and evolution of the genetic code. Transcription 2018, 10.1080/21541264.2018.1429837, doi:10.1080/21541264.2018.1429837.

Kim, Y.; Kowiatek, B.; Opron, K.; Burton, Z.F. Type-II tRNAs and Evolution of Translation Systems and the Genetic Code. Int J Mol Sci 2018, 19, doi:10.3390/ijms19103275.

Pak, D.; Root-Bernstein, R.; Burton, Z.F. tRNA structure and evolution and standardization to the three nucleotide genetic code. Transcription 2017, 8, 205-219, doi:10.1080/21541264.2017.1318811.

We have added in Supplementary Information the ancestral sequences of tRNA with which the PTC was formed. We also dedicated some paragraphs about the method used to obtain the PTC. Our work is focused on the origin of PTC and its current diversity. The references suggested by the reviewer are related to the origin of tRNAs. These works do not have a direct bearing with our work.

Round 2

Reviewer 2 Report

The authors have tried to improve the paper. I have no further suggestions.